# Intercellular propagation of extracellular signal-regulated kinase activation revealed by in vivo imaging of mouse skin

**Toru Hiratsuka[1], Yoshihisa Fujita[1], Honda Naoki[2], Kazuhiro Aoki[2], Yuji Kamioka[1], Michiyuki Matsuda[1,3]***

[1]Department of Pathology and Biology of Diseases, Graduate School of Medicine, Kyoto University, Kyoto, Japan; [2]Imaging Platform for Spatio-Temporal Information, Graduate School of Medicine, Kyoto University, Kyoto, Japan; [3]Laboratory of Bioimaging and Cell Signaling, Kyoto University, Kyoto, Japan

**Abstract** Extracellular signal-regulated kinase (ERK) is a key effector of many growth signalling pathways. In this study, we visualise epidermal ERK activity in living mice using an ERK FRET biosensor. Under steady-state conditions, the epidermis occasionally revealed bursts of ERK activation patterns where ERK activity radially propagated from cell to cell. The frequency of this spatial propagation of radial ERK activity distribution (SPREAD) correlated with the rate of epidermal cell division. SPREADs and proliferation were stimulated by 12-O-tetradecanoylphorbol 13-acetate (TPA) in a manner dependent on EGF receptors and their cognate ligands. At the wounded skin, ERK activation propagated as trigger wave in parallel to the wound edge, suggesting that ERK activation propagation can be superimposed. Furthermore, by visualising the cell cycle, we found that SPREADs were associated with G2/M cell cycle progression. Our results provide new insights into how cell proliferation and transient ERK activity are synchronised in a living tissue.

**\*For correspondence:** matsuda. michiyuki.2c@kyoto-u.ac.jp

**Competing interests:** The authors declare that no competing interests exist.

**Reviewing editor**: Roger Davis, University of Massachusetts Medical School, United States

## Introduction

Exquisite control of extracellular signal-regulated kinase (ERK) MAPK signalling is required for homeostasis in the epidermis and other tissues (*Pouysségur et al., 2002*; *Kholodenko, 2006*; *Khavari and Rinn, 2007*; *Roberts and Der, 2007*). ERK activation is triggered by the activation of epidermal growth factor receptors (EGFRs) through binding to their cognate ligands (*Yuspa, 1994*; *Sibilia et al., 2007*; *Schneider et al., 2008*). Precursors of EGFR ligands including EGF, transforming growth factor α (TGFα), and heparin binding EGF-like growth factor (HB-EGF), are generated as membrane-bound proteins and are released through cleavage by matrix metalloproteinases (MMPs) including ADAM17 (*Massagué and Pandiella, 1993*; *Sahin et al., 2004*). The critical role of MMP-mediated release of EGFR ligands is corroborated by the close resemblance of the phenotypes among knockout mice of TGFα, ADAM17, and EGFR (*Luetteke et al., 1993*; *Peschon et al., 1998*; *Franzke et al., 2012*). However, little is known about how such cell-to-cell communication leading to activation of the Ras/ERK pathway is dynamically regulated in vivo.

To address this issue, we live imaged transgenic (Eisuke) mice expressing a Förster resonance energy transfer (FRET) biosensor for ERK in order to visualise ERK activity in living epidermis under two photon excitation microscopy (*Kamioka et al., 2012*). During long-time time lapse imaging of the steady-state mouse skin, we noticed the epidermis occasionally appear with bursts of ERK activation patterns, where ERK activation is propagated from cell to cell in a radial wave. We named this new phenomenon as Spatial Propagation of Radial ERK Activity Distribution (SPREAD) and investigated

**eLife digest** Our skin is our largest organ; it provides a barrier that protects the underlying tissues and internal organs from the external environment and acts as one of our first lines of defense against infection. Both of these roles subject the skin to wear and tear and so it must constantly create new skin cells to replace those lost or damaged. However, if this renewal process goes awry it can lead to excessive cell growth or skin cancer. To avoid this, cells tightly regulate the pathways that stimulate skin renewal.

Skin renewal involves growth signals activating an enzyme called ERK. When and where the ERK enzyme is activated is normally tightly regulated, and many kinds of cancer have been linked to ERK becoming active at the wrong time or in the wrong place. Despite the importance of ERK in skin cells, a number of technical challenges have made it difficult to study how these signals are passed from cell to cell.

Hiratsuka et al. have now examined genetically altered mice that produce a fluorescent sensor molecule that makes it possible to see ERK activity in living skin cells. The skin of anesthetized mice was observed under a microscope, and time-lapse videos revealed occasional 'firework-like' bursts of ERK activity. At first the ERK enzyme was active in a small cluster of skin cells, then ERK activity was seen in the surrounding cells—appearing to spread outwards over the course of several minutes—before the activity stopped. Hiratsuka et al. named this pattern of activity a 'Spatial Propagation of Radial ERK Activity Distribution', or SPREAD for short.

By studying SPREADs in the skin on the ears and the back of these mice, Hiratsuka et al. learned that these bursts of ERK activity coincided with skin cell growth; the bursts happened more frequently in the areas where the skin cells were dividing. Applying a chemical that stimulates cell division to the skin of the mice triggered more bursts of ERK activity; whereas fewer bursts were observed if Hiratsuka et al. used other chemicals to block the activity of some of the signaling proteins that work upstream of ERK.

Further experiments suggested that SPREADs encourage cells to progress through the cycle of events that leads a cell to divide; blocking these bursts caused the cell to pause at the stage just before it would normally divide. Hiratsuka et al. also observed similar patterns of ERK activity moving out like waves from the edges of skin wounds. Further research using similar methods will reveal how growth signals are triggered and propagated in healthy and diseased tissues, not only in the skin but also other organs such as the liver, intestine, and muscles.

characteristics, mechanisms, and roles of SPREAD. Single-cell analysis of SPREAD demonstrated that ERK activation in SPREAD originated from a cluster of a few cells and propagated up to about 50 μm with an average velocity of 1.5 μm/min. The amplitude of ERK activation and efficiency in propagation gradually decreased until SPREAD disappeared. Interestingly, the frequency of SPREAD was spatio-temporally associated with that of cell division. SPREADs were significantly stimulated by topical treatment of mitotic stimulation, 12-O-tetradecanoylphorbol 13-acetate (TPA). The induction was dependent on EGF receptors and the production of their cognate ligands by MMPs. The role of SPREAD in cell cycle progression was studied with Fucci mice, in which G0/G1 cells and S/G2/M cells are visualised by the expression of mKO2-Cdt1 and mAG-Geminin, respectively (*Sakaue-Sawano et al., 2008*). Inhibition of SPREAD in Fucci mice led to significant delay of entry to G0/G1 phase from S/G2/M phase under TPA treatment, suggesting the role of SPREAD in G2/M progression of cell cycle.

We also imaged ERK activity of ear skin subjected to epithelial wounding. Interestingly, wounded skin revealed another type of ERK activation propagation pattern, where waves of ERK activation propagated in parallel with the wound edge. Single cell analysis revealed similarity and difference between the wound-induced waves and SPREAD. While the velocity of ERK activity propagation in wound-induced waves was similar to that of SPREAD, the ERK activation waves from the wound edge was relatively maintained during propagation during propagation.

In this study, we propose how dynamically growth signal is regulated in steady-state and wounded skin in a living tissue by showing two types of novel ERK activation patterns.

## Results

### In vivo imaging of ERK activity at single cell resolution

We recently reported Eisuke mice, which can visualise ERK activity at a single cell resolution by the expression of a Förster resonance energy transfer (FRET) biosensor for ERK (*Kamioka et al., 2012*). In Eisuke mice used in this study, the nuclear ERK biosensor is expressed ubiquitously under the control of the CAG promoter (*Figure 1A*). We imaged mainly ear epidermis of anaesthetised mice by two-photon excitation microscopy. The ear skin was stabilized between thermo-conductive silicon gum sheet and a cover glass to avoid movement derived from heart pulses and respirations (*Figure 1B*). Images were acquired every 5 min with a viewfield of 0.213 mm$^2$ in 2–3 μm steps using a 25X/1.05 water-immersion objective lens (XLPLN 25XWMP; Olympus, Tokyo, Japan). In our imaging settings for two photon microscopy, skin cells were clearly imaged from hair follicles to the outer epidermal layers (stratum corneum). The level of FRET was analysed by the FRET/CFP ratio image and shown in intensity modulated display mode; eight colours from red to blue are used to represent the FRET/CFP ratio and the 32 grades of colour intensity are used to represent the signal intensity of the CFP image. The warm and cold colours indicate high and low FRET levels, namely ERK activity, respectively. Epidermal layers and dermis were clearly discerned by cellular size and collagen fibres visualised by second harmonic generation (SHG) microscopy (*Figure 1C–D*). Autofluorescence was negligible in the skin of control mice on the same background (FVB/N) (*Figure 1—figure supplement 1*). Backskin of Eisuke mice was successfully visualised by a similar method as ear skin (*Figure 1—figure supplement 2*).

### Radial propagations of ERK activation in steady-state epidermis

During long-term time lapse imaging for more than 200 hr in total, we noticed firework-like bursts of ERK activation and propagation in steady-state ear skin (*Figure 2A* and *Video 1*). The ERK activation pattern originated from a few cells and radially propagated mainly in the basal layer of epidermis and to a lesser extent in the suprabasal layer (*Figure 2B*). We named this focal and transient ERK activation as 'Spatial Propagation of Radial ERK Activity Distribution' (SPREAD). It is also noteworthy that SPREADs were also observed in the epidermis of backskin, indicating that SPREADs are generally observed in mouse epidermis (*Figure 2—figure supplement 1*).

### Single cell analysis of SPREAD

Detailed analysis of SPREADs was performed with a time-lapse image acquired at shorter intervals (90 s). Among eight SPREADs (*Figure 3—figure supplement 1A*), 'SPREAD 6' is shown as the example of analysis (*Figure 3*). The centres of SPREADs were tentatively determined by visual inspections, which were optimised in later analyses. A square region was set to encompass the entire area of each SPREAD and individual cells were recognised by a segmentation programme (*Figure 3A* and *Figure 3—figure supplement 1A*). Then, we attempted to fit the FRET/CFP value of each cell to a flat line of the average FRET/CFP value during the analysed period. If the line fitting was successful, the cell of interest was classified as a non-responder, otherwise the cell was fit to one to three sine curve(s) and was classified as a responder (*Figure 3B–C* and *Figure 3—figure supplement 1B*). Based on the fitted data, the peak time and the amplitude of ERK activation were determined for each cell. The mean duration of the activation wave was 47 ± 15 min (n = 582). By plotting the peak time against the distance from the centre of the SPREAD, the velocity of ERK propagation was determined as 1.93 μm/min (*Figure 3D*). The mean velocity of the eight SPREADs was 1.5 ± 0.3 μm/min (*Figure 3—figure supplement 1C*). We also applied the square root fitting, which yielded similar coefficient determination values; therefore, we could not conclude whether the wave slows down during propagation (*Figure 3—figure supplement 2*). The amplitude of ERK activation of each responder cell decreased inversely with the distance of the cell from the origin of the SPREAD (*Figure 3E* and *Figure 3—figure supplement 1D*). The number of responders reached the zenith around 5 min after the initiation of SPREAD and subsequently decreased (*Figure 3F* and *Figure 3—figure supplement 1E*). The radius of SPREAD was determined as the distance from the SPREAD centre to the inflection point of the plotting of the fraction of responder cells. In the current analysis set-up, the duration and the radius of SPREADs were 30.5 ± 5.6 min and 47.7 ± 10.2 μm, respectively (*Figure 3G* and *Figure 3—figure supplement 1F*). We performed this analysis in 8 SPREADs (*Figure 3—figure supplement 1*) and found that the size of SPREAD is positively correlated with the Max ERK activity in SPREAD and the wave velocity (*Figure 3H–I*).

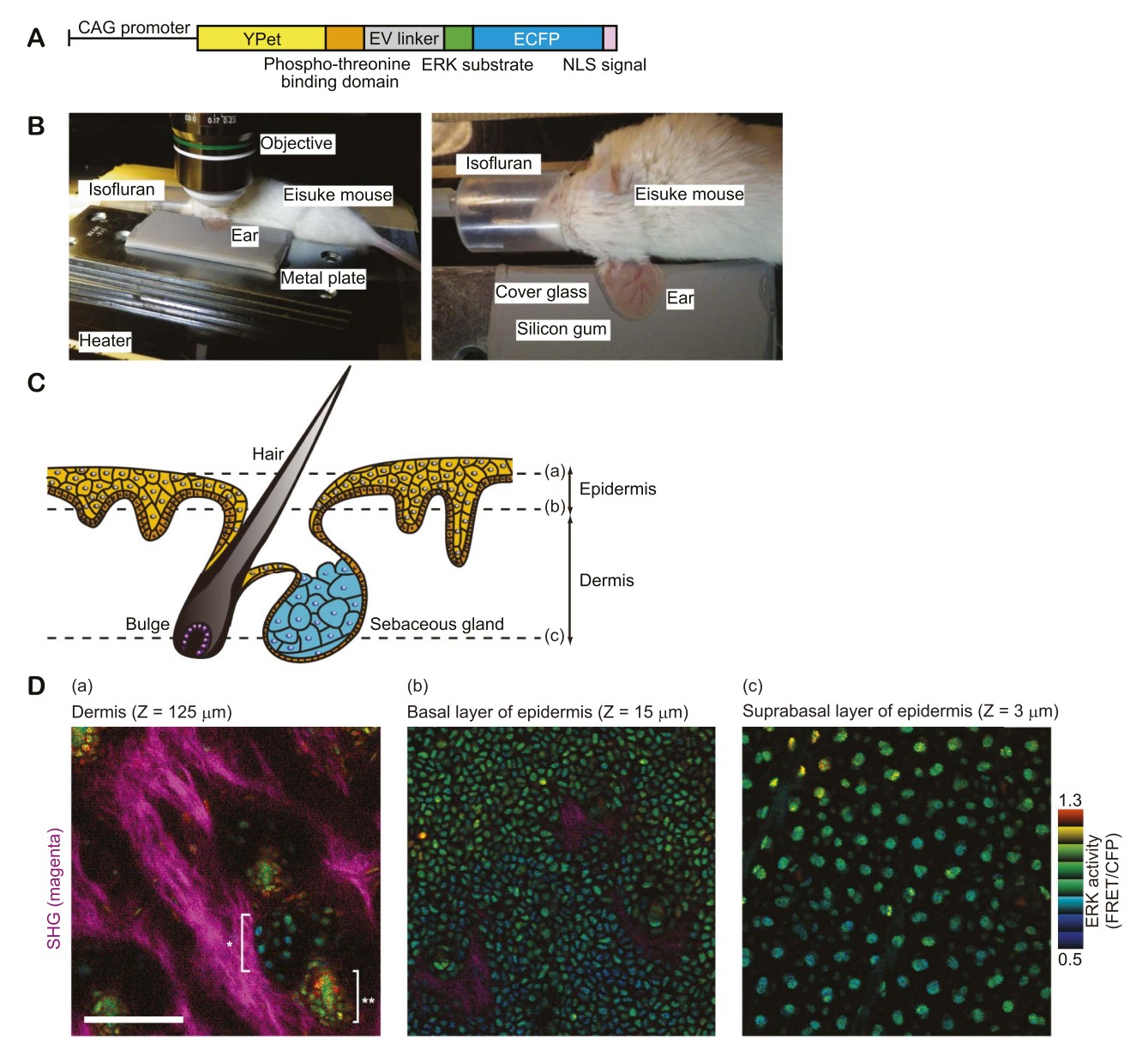

**Figure 1**. In vivo imaging of ERK activity in ear skin. (**A**) Structure of ERK biosensor (EKAREV-NLS) expressed in Eisuke mice. (**B**) Experimental set-up of in vivo imaging of ear skin with a two-photon microscope. (**C**) Schematic of skin structure. (**D**) ERK activity maps in three different skin layers indicated in (**C**). ERK activity in the nuclei is represented by intensity modulated display (IMD) mode. The warm and cold colours indicate high and low ERK activities, respectively. Collagen fibres (magenta) were detected by second harmonic generation microscopy. *indicates sebaceous gland and **indicates hair bulge. Scale bar, 100 μm.

The following figure supplements are available for figure 1:

**Figure supplement 1**. Autofluorescence in background FVB/N mouse.

**Figure supplement 2**. In vivo imaging of ERK activity in backskin.

Previously, Albeck et al. and we reported that ERK can be spontaneously and transiently activated in culture cells and we reported that the ERK activity pulse in a single cell could be propagated to the neighbouring cells (*Albeck et al., 2013*; *Aoki et al., 2013*). There are some differences between the propagation of ERK activation in vitro and in vivo, that is, SPREADs. First, the amplitudes of ERK

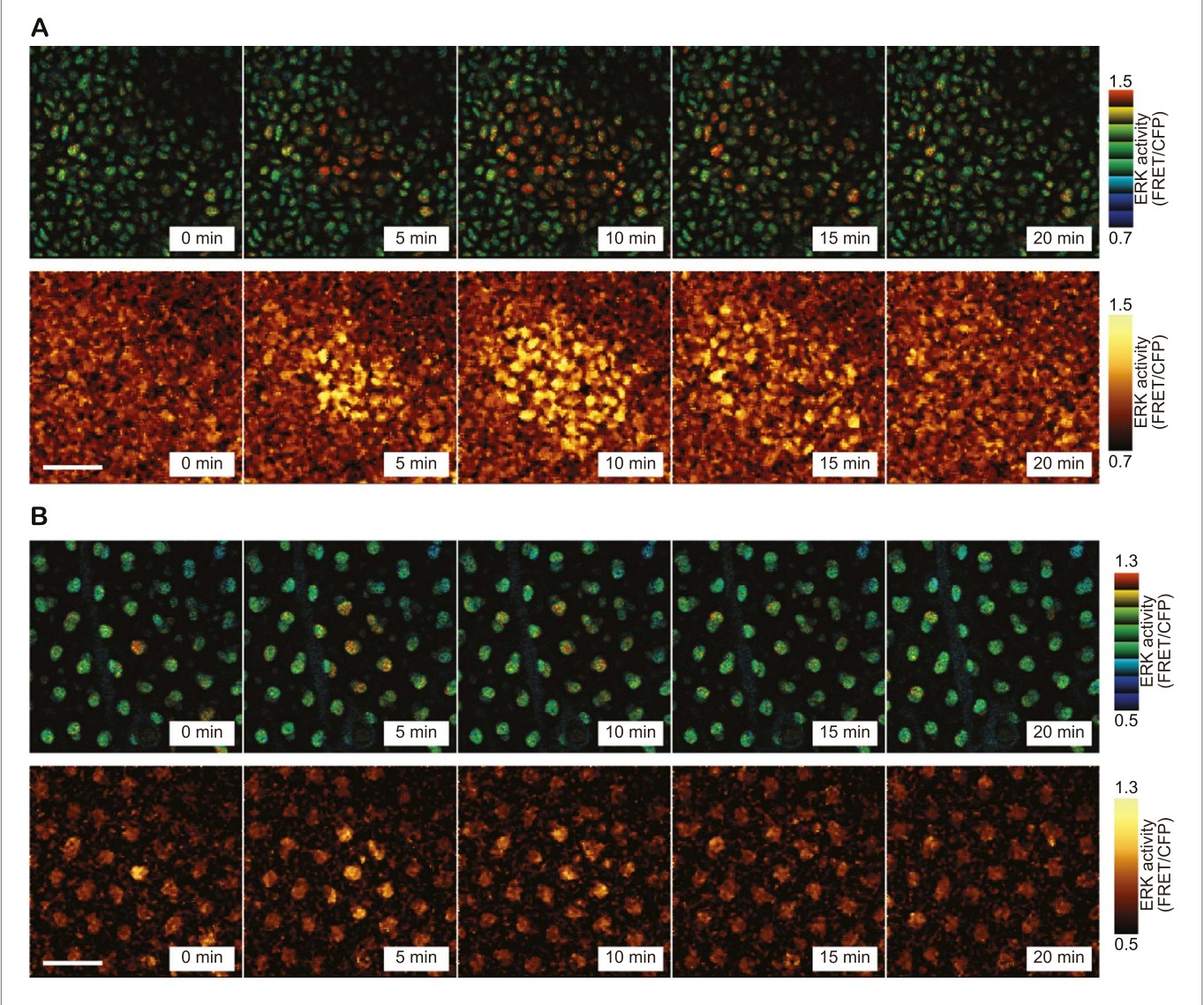

**Figure 2**. SPREAD in ear epidermis. (**A**) Representative time-lapse images of SPREAD in the basal layer of ear epidermis. FRET/CFP ratio is represented in IMD mode (upper panels) and gold pseudo-colours (lower panels). The images are cropped from *Video 1*. Scale bar, 30 µm. (**B**) Suprabasal layer of ear epidermis in the same area shown in (**A**). FRET/CFP ratio is represented in IMD mode (upper panels) and gold pseudo-colours (lower panels). Scale bar, 30 µm.

The following figure supplement is available for figure 2:

**Figure supplement 1**. SPREAD in backskin epidermis.

activation in vitro are relatively constant during propagation, whereas in SPREADs the amplitudes of ERK activation decay during propagation. Second, ERK activation was propagated initially to almost all neighbouring cells in SPREADs, whereas ERK activity was propagated stochastically in vitro to the neighbouring cells. Third, SPREAD seems to emerge from a small fraction of cells, which is not the case in vitro.

## The frequency of SPREAD positively correlates with that of cell division

Interestingly, the frequency of SPREAD was not constant. Instead, there were periods when SPREADs are more frequently observed than other periods (*Figure 4A–B* and *Figure 4—figure supplement 1A*). Furthermore, we counted cell divisions by visual inspection of time lapse images and found that cell divisions were also more abundantly observed in the periods with frequent SPREADs than those with

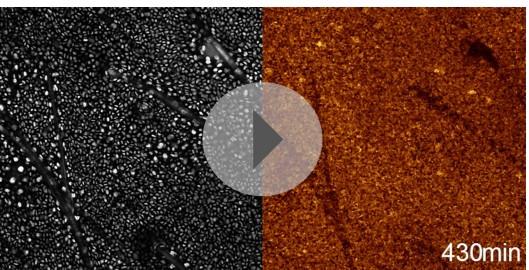

430min

**Video 1**. SPREADs detected by in vivo imaging of an Eisuke mouse. A time-lapse video of steady-state basal epidermis of an Eisuke mouse for 12 hr. The skin shows localized ERK activations that concentrically propagate. The FRET/CFP ratio is shown by gold pseudo-colours (right). Images were acquired every 5 min. The area of the viewfield is 0.213 mm². HF: hair follicle.

less frequent SPREADs. For example, while mice #102, #107, #112, #115, and #117 showed frequent SPREADs and cell divisions, mice #113, #114, and #120 showed much less SPREADs and cell divisions (*Figure 4A–B*, *Figure 4—figure supplement 1A*). The cause of the heterogeneity of SPREADs and cell division among the mice is not clear. We failed to find any relationship between the SPREAD frequency and sex or age of the mice (*Figure 4—figure supplement 1A*). Mice with frequent SPREADs (mice #102, #107, #112, #115, and #117) showed higher basal ERK activity than those with infrequent SPREADs (mice #113, #114, and #120); however, the difference was not statistically significant (*Figure 4—figure supplement 1B*). Improvement of long-term imaging system and accumulation of data are required to answer this question. Notably, the period with frequent SPREADs and cell divisions was also observed in the backskin (*Figure 4—figure supplement 2*), indicating that SPREAD is not limited to the ear skin.

For quantification of the spatio-temporal relationship between SPREAD and cell division in the eleven ear skin imaging data set, SPREAD area (SA) was defined as the area and time within 100 µm from the origins and within 1 hr of the onset of the SPREADs, while the excluded area was defined as Non-SPREAD area (Non-SA). The size of SA was inferred from the data shown in *Figure 3G*. Cell division in SAs was twice as frequent as that in Non-SAs (*Figure 4C*). It is noteworthy that about 50% of SPREADs emerged from within 50 µm of hair follicles, which occupied 26% of ear skin on average, indicating that SPREADs were initiated preferentially in, but not necessarily restricted to the para-hair follicle regions. To examine statistically whether there are hot spots for SPREAD generation, we tested the null hypothesis that the origins of SPREADs are distributed randomly. In this test, we adopted, as a test statistic, average distance of from the centre of each SPREAD to the nearest neighbour. Then, we generated the histogram on the null hypothesis by 100,000 trials of Monte Carlo simulations. We found that actual value of the test statistic was significantly lower than the expectation based on the histogram (p < 0.01 in one-sided test), indicating that the SPREADs were significantly clustered (*Figure 4—figure supplement 3*). Furthermore, SPREADs often emerged repeatedly from the same spot (*Figure 4D*). These results suggest that SPREAD may have a relationship to follicular and interfollicular stem cells (*Jones et al., 1995*; *Arwert et al., 2012*; *Blanpain and Fuchs, 2014*).

## SPREADs induced by TPA treatment

The association of SPREADs with cell divisions prompted us to examine the effect of a mitotic stimulant, 12-O-tetradecanoylphorbol 13-acetate (TPA) (*Figure 5A*). Upon topical TPA treatment, ERK activity was gradually stimulated, reaching a plateau approximately 6 hr later. ERK activity remained high for more than 24 hr and gradually decreased to the basal state by 48 hr (*Figure 5B*). Unexpectedly, we failed to observe an increase in the frequencies of SPREAD and cell division following a single TPA treatment (*Figure 5G–H*, TPA1). However, when we applied a second dose of TPA, according to the established protocol for tumorigenesis (*Abel et al., 2009*), the frequencies of SPREAD and cell division were significantly increased 10–18 hr later (*Figure 5C–F*, and *Video 2*). This observation suggests that the increase in SPREAD frequency requires the induction and/or synchronization of cells in the replicating cycle upon the second TPA treatment. It has been shown that a single EGF pulse induces cell cycle arrest in the G1 phase by p53-mediated mechanisms and two EGF pulses promote cell cycle progression in mammary epithelial cells (*Zwang et al., 2011*). Although the interval between the two EGF stimulations is only 7 hr in this in vitro study, similar mechanism may operate in the mouse epidermal cells, where the cell cycle period is markedly longer than in tissue mammary epithelial cells.

## The involvement of EGFRs and their ligands in SPREAD emergence

We previously showed in vitro that lateral propagation of ERK activity pulses was dependent on MMPs and EGFR activity, indicating that cleavage of pro-EGFR-ligands is critical for the signal propagation

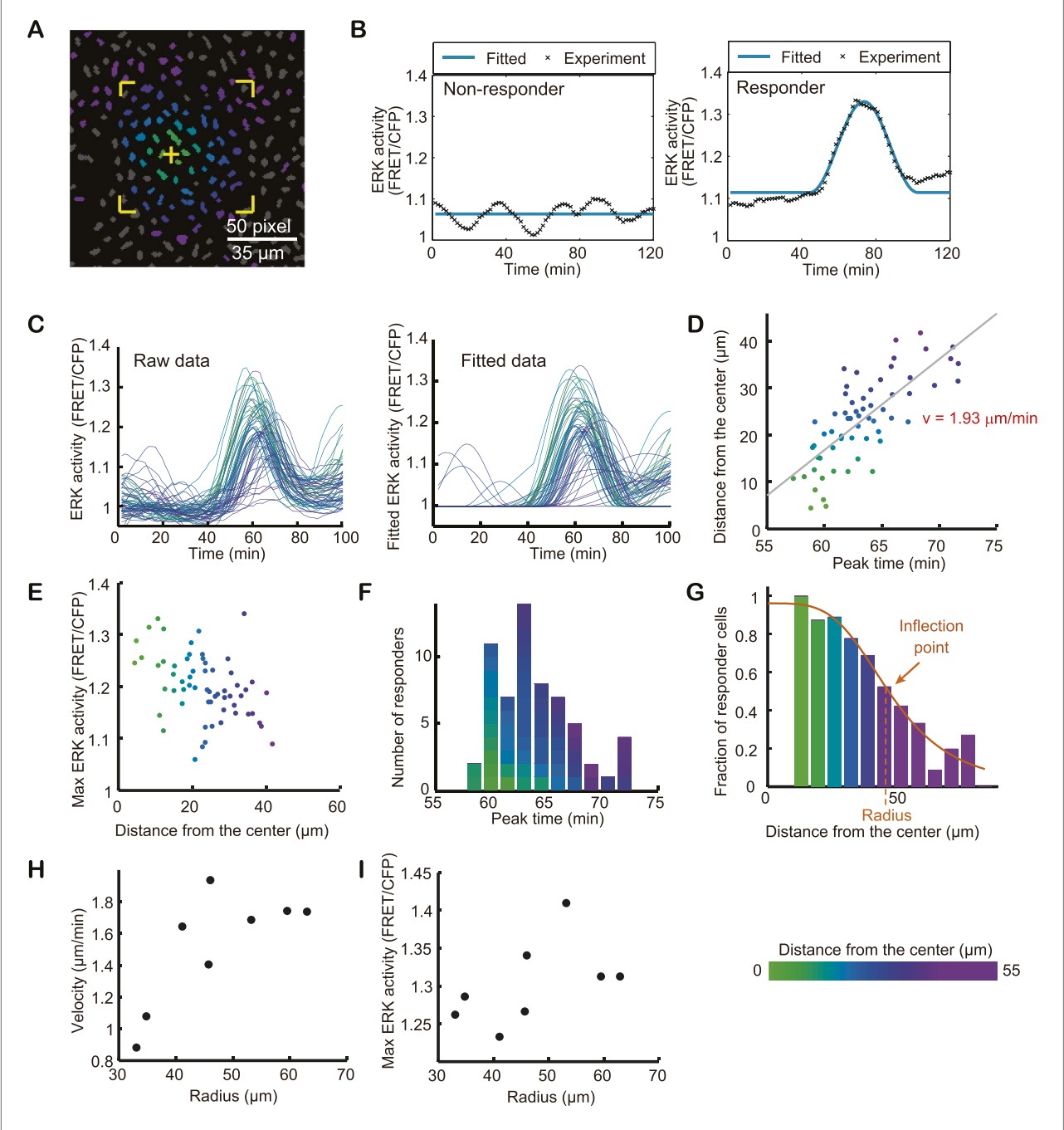

**Figure 3.** Single cell analysis of SPREAD. Detailed analysis of SPREADs was performed with a time-lapse images acquired at 90-s-interval. (**A**) Binarized CFP image of the SPREAD area. The nucleus of each cell was identified and classified as a non-responder (grey) or a responder (coloured) as described in the 'Materials and methods' section. Cells within the region marked by yellow corners were analysed in (**C–F**). All cells in the viewfield were analysed in (**G**). The yellow cross indicates the origin of the SPREAD. (**B**) Time series of raw ERK activity (FRET/CFP) (black dotted lines) were approximated with a flat line (left) or sine curve(s) (right). (**C**) Raw (left) and fitted (right) time series of ERK activity (FRET/CFP) of responders. (**D**) The distance from the centre of SPREAD to each cell was plotted against the peak time to determine the velocity of ERK propagation. (**E**) The Max ERK activity of each cell plotted against the distance from the centre of the SPREAD. (**F**) Peak time histogram of responder cells. (**G**) The fraction of responder cells in each class of distance from the centre. Cells in the most central class were not analysed because there were only three cells. (**H** and **I**) Eight SPREADs were analysed to examine the correlation between the radius with the velocity of ERK activation propagation (**H**) and the Max amplitude of the ERK activity (**I**) of each SPREAD.

*Figure 3. Continued on next page*

*Figure 3. Continued*

The following figure supplements are available for figure 3:

**Figure supplement 1**. Single cell analysis of individual SPREADs.

**Figure supplement 2**. Straight and square root line fitting of distance-time plot of SPREAD.

(*Aoki et al., 2013*). To examine if a similar mechanism operates in SPREADs, Eisuke mice were topically pre-treated with TAPI-1, an MMP inhibitor, or PD153035, an EGFR inhibitor, before the second TPA treatment in the double-TPA stimulation protocol (*Figure 5C*). Both inhibitors significantly decreased the frequency of SPREADs as well as that of cell division (*Figure 5G–H*). This is in line with the studies showing that TPA drives cleavage of pro-EGFR-ligands in a MMPs-dependent manner (*Dethlefsen et al., 1998*; *Izumi et al., 1998*; *Le Gall et al., 2003*). These observations suggest that

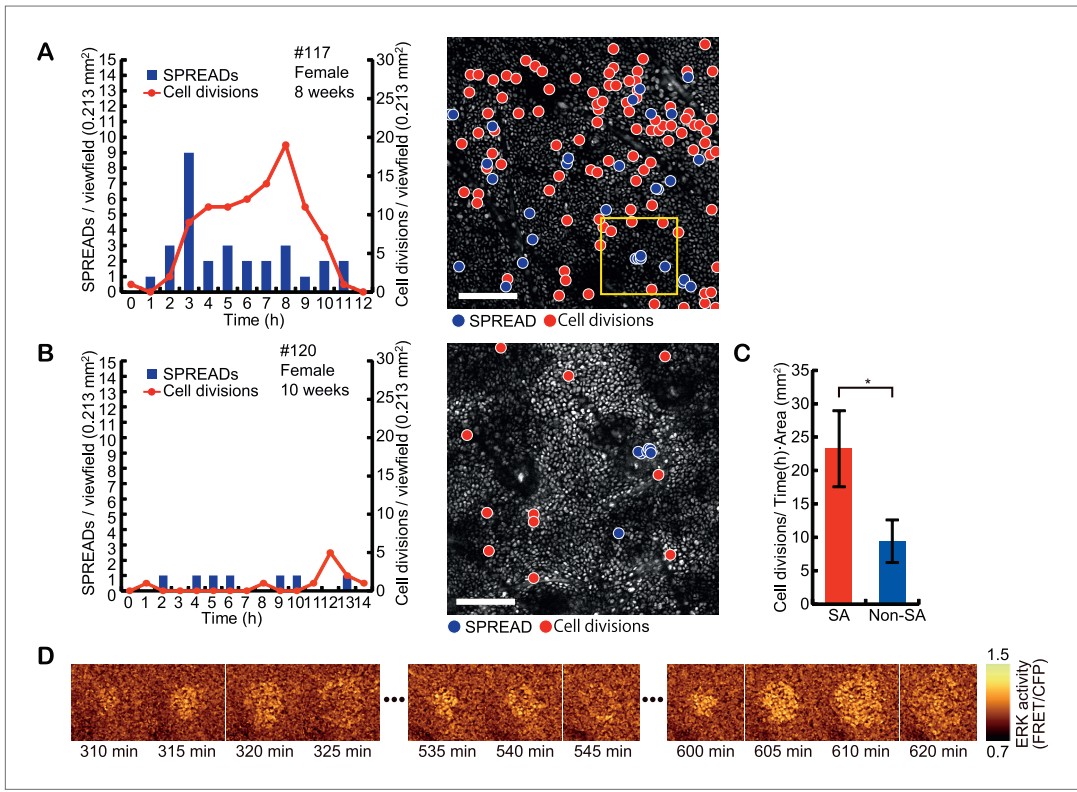

**Figure 4**. Spatio-temporal association of SPREADs with cell divisions. (**A** and **B**) Left panels show frequencies of SPREADs (blue) and cell divisions (red) during periods with frequent SPREADs (**A**) and less frequent SPREADs (**B**). Right panels show mapping of the centres of SPREADs (blue circle) and cell divisions (red circle) observed during each imaging period. Shown are single frames of CFP images of a 0.213 mm² viewfield. Individual mouse identification numbers, sex and age are shown on the top of the left panels. Scale bar, 100 μm. (**C**) The frequency of cell division in and out of the SPREAD area. SPREAD area was defined as the area within 100 μm from the origin and within 1 hr of the onset of the SPREAD. *p < 0.05 (paired Student's *t* test). (**D**) Time-lapse images of the yellow square region in (**A**), showing three SPREADs emerging from the same spot.

The following figure supplements are available for figure 4:

**Figure supplement 1**. Frequencies of SPREAD and cell division in the steady-state ear skin of eleven mice.

**Figure supplement 2**. Steady-state backskin with frequent SPREADs and cell divisions.

**Figure supplement 3**. Monte Carlo simulation of SPREAD distribution.

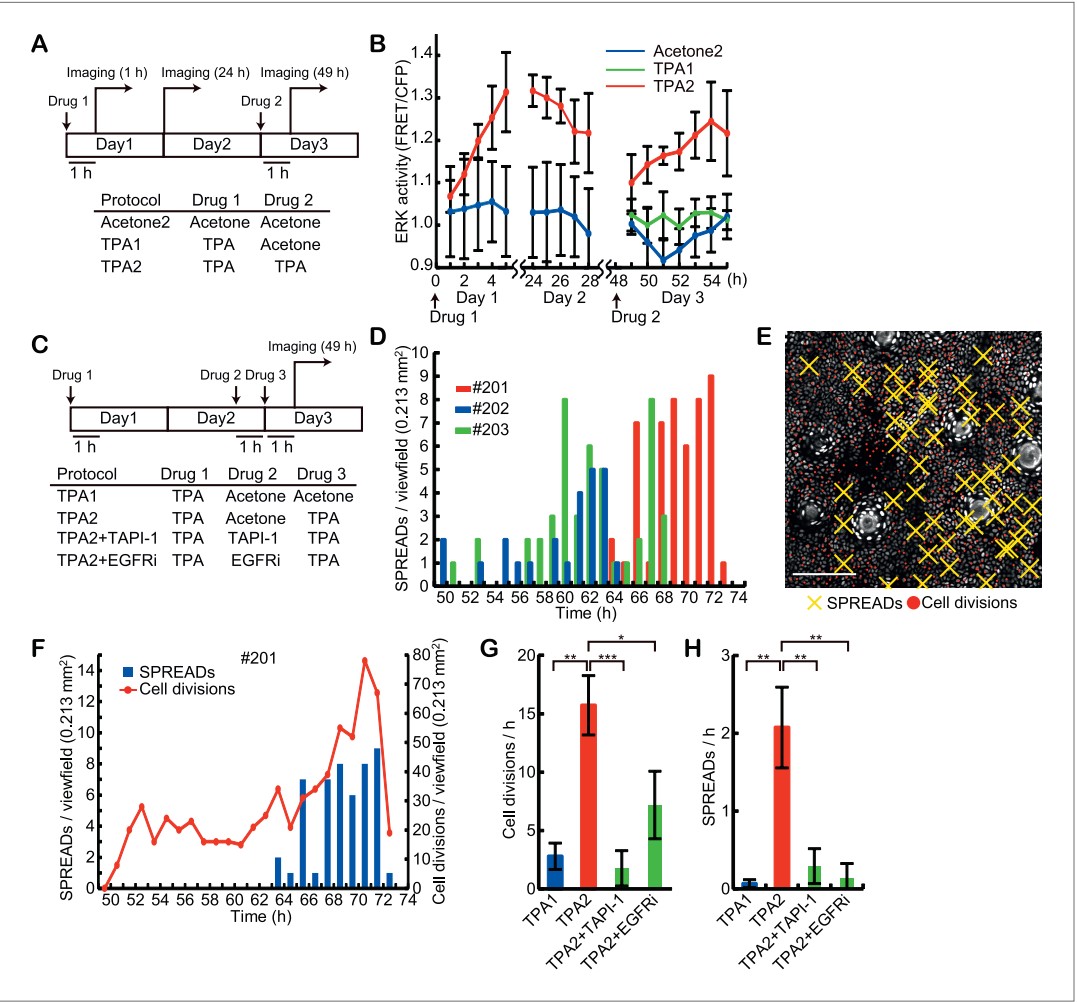

**Figure 5**. Induction of SPREADs by the double TPA treatments. (**A**) Eisuke mice were subjected to topical application of 20 μl acetone with or without 0.5 nanomole TPA and imaged according to the protocols. Images were acquired every 1 hr during 5- to –7-hr imaging periods. Mice were returned to the cage after the imaging and the same area of the skin was revisited on the next day. (**B**) ERK activity in the basal epidermal cells was monitored in FVB[EKAREV–NES] mice. Average FRET/CFP ratio and S.E.M. of three mice are shown for each. Error bars indicate S.E.M. of the three mice. (**C**) Protocols for the double TPA application in the presence of inhibitors. Drugs were used as follows: 0.5 nanomole TPA, 207 nanomole PD0329105, 2.0 nanomole TAPI-1, and 0.2 nanomole PD153035 in 20 μl acetone, or vehicle alone. (**D**) A histogram of SPREADs of the three mice, #201, #202, and #203, which were treated twice with TPA2 in (**C**). (**E**) A map of SPREADs (yellow cross) and cell divisions (red dot) observed during 48–72 hr in the mouse #201. (**F**) The numbers of SPREADs and cell divisions observed in each period. Data are from mouse #201. (**G** and **H**) Numbers of cell divisions (**G**) and SPREADs (**H**) observed in a 0.213 mm² viewfield per hour during the observation. Average and S.E.M. are shown. At least three mice were analysed for each protocol. *$p < 0.05$; **$p < 0.01$; ***$p < 0.001$ (paired Student's *t* test).

the propagation of ERK activation in SPREAD is mediated by EGFR-ligands released by MMPs as was the case in vitro.

## The role of SPREAD in cell cycle progression

For detailed analysis of the role of SPREADs in cell cycle progression, we employed Fucci mice, in which G0/G1 cells and S/G2/M cells are visualised by the expression of mKO2-Cdt1 and mAG-Geminin, respectively (*Sakaue-Sawano et al., 2008*). To examine the spatial and temporal variation in the proportion of cycling cells, images of 120 viewfields were acquired at various locations and times from 12 mice with ages ranging between 8 to 20 weeks. In each viewfield, 723 cells on average were classified into G0/G1 or S/G2/M phase based on the fluorescence intensities of mKO2-Cdt1 and

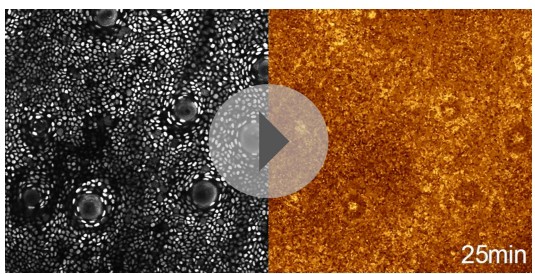

25min

**Video 2**. SPREAD induction by topical applications of TPA. A Time-lapse video of an Eisuke mouse topically treated with TPA twice as described in **Figure 5C** (TPA2). The video starts from 12 hr after the second TPA treatment (60 hr after the first TPA treatment). The video length is about 12 hr (735 min). SPREADs appeared from about 300 min indicated in the video. The FRET/CFP ratio is shown by gold pseudo-colours (right). Images were acquired every 5 min. The area of the viewfield is 0.213 mm². HF: hair follicle.

mAG-Geminin (**Figure 6A**). While the average proportion of S/G2/M cells was 12.2%, 2.5% of the viewfields showed more than 50% of S/G2/M cells, which might correspond to the periods of frequent SPREADs (**Figure 6A**, double-asterisk, and **Figure 6B**, right panel). We then performed the double TPA treatment experiments with Fucci mice. In a preliminary experiment, we found that the fraction of S/G2/M cells after the second TPA application was increased from 61% to 95% by reducing the interval from 48 hr to 24 hr. Because high synchronization of cell cycle is helpful to examine the effect of inhibitors clearly, we applied this modified protocol to the following experiments (**Figure 6C**). Single TPA treatment alone drove a maximum of ca. 40% cells into the S/G2/M phase at 30 hr after treatment (**Figure 6D**, black line), followed by gradual decrease of the proportion of the S/G2/M phase. In contrast, the second TPA treatment increased the proportion of S/G2/M cells up to 95% within 12 hr (36 hr after the first TPA), which decreased to about 20% at

58 hr after the first TPA (**Figure 6D**, red line). PD0325901, a MEK inhibitor, cancelled the effect of the second TPA treatment (**Figure 6D**, green line). In contrast to the MEK inhibitor, TAPI-1 did not suppress the increase in the S/G2/M cells (**Figure 6D**, blue line). Surprisingly and importantly, however, the percentage of S/G2/M cells remained very high until 58 hr, indicating that TAPI-1 inhibited exit from the S/G2/M phase and entry to the G0/G1 phase (**Figure 6D–E**). Together with the reduced frequency of cell division in the TAPI-1-treated Eisuke mice (**Figure 5G**), this observation suggests that SPREADs play an important role in G2/M progression.

## ERK activation propagation during wound healing

To study whether SPREADs are induced in a pathologically growth-accelerated condition, we followed ERK activity during wound healing of ear skin. SPREADs were frequently observed in regions ca. 200 µm from the wound edge (**Figure 7A–D**, **Video 3**). The induction of SPREADs during wound healing again raises the possibility that stem cell activation is involved in their emergence (**Jones et al., 1995**; **Arwert et al., 2012**; **Blanpain and Fuchs, 2014**). As with TPA-induced SPREADs, both SPREADs and cell divisions during wound healing were significantly reduced by TAPI-1 or PD153035 treatment (**Figure 7E,F**). Interestingly, in regions within about ca. 100 µm from the wounded edge, we found another type of ERK activation propagation pattern (**Figure 7C** and **Video 4**). Here, repetitive waves of ERK activation parallel to the wound edge emerged and propagated to cells ca. 100 µm from the wound edge (**Figure 7C**). The ERK activation waves from the wound edge were suppressed by TAPI-1 treatment, suggesting the involvement of the similar mechanisms as that of SPREAD (**Video 5**). Single-cell analysis of the ERK activation waves from the wound was performed with segmentation and sine curve fittings as performed for SPREADs in **Figure 3** (**Figure 8A–C**). The velocity of ERK activity propagation was determined as 1.4 µm/min, which is similar to that in SPREAD (**Figure 8D**). In contrast to ERK propagation in SPREAD, ERK activation waves from the wound edge maintained high ERK activity and propagation efficiency (**Figure 8E–F**), suggesting that ERK activation propagation can be synchronized and superimposed to generate trigger waves if a number of ERK activation pulses are evoked in close temporal and spatial proximity. These similarities and differences between SPREADs and ERK activation waves from the wound edge suggest the possible modification of the amplitude and efficiency in propagation depending on the surrounding environment of the cells.

## Discussion

Cell proliferation at proper time and place is essential for tissue maintenance in multicellular organisms. In steady-state skin, epidermal cells undergo continuous turnover through autonomous activation of growth signal pathway triggered by the interaction of EGFRs and their cognate receptors

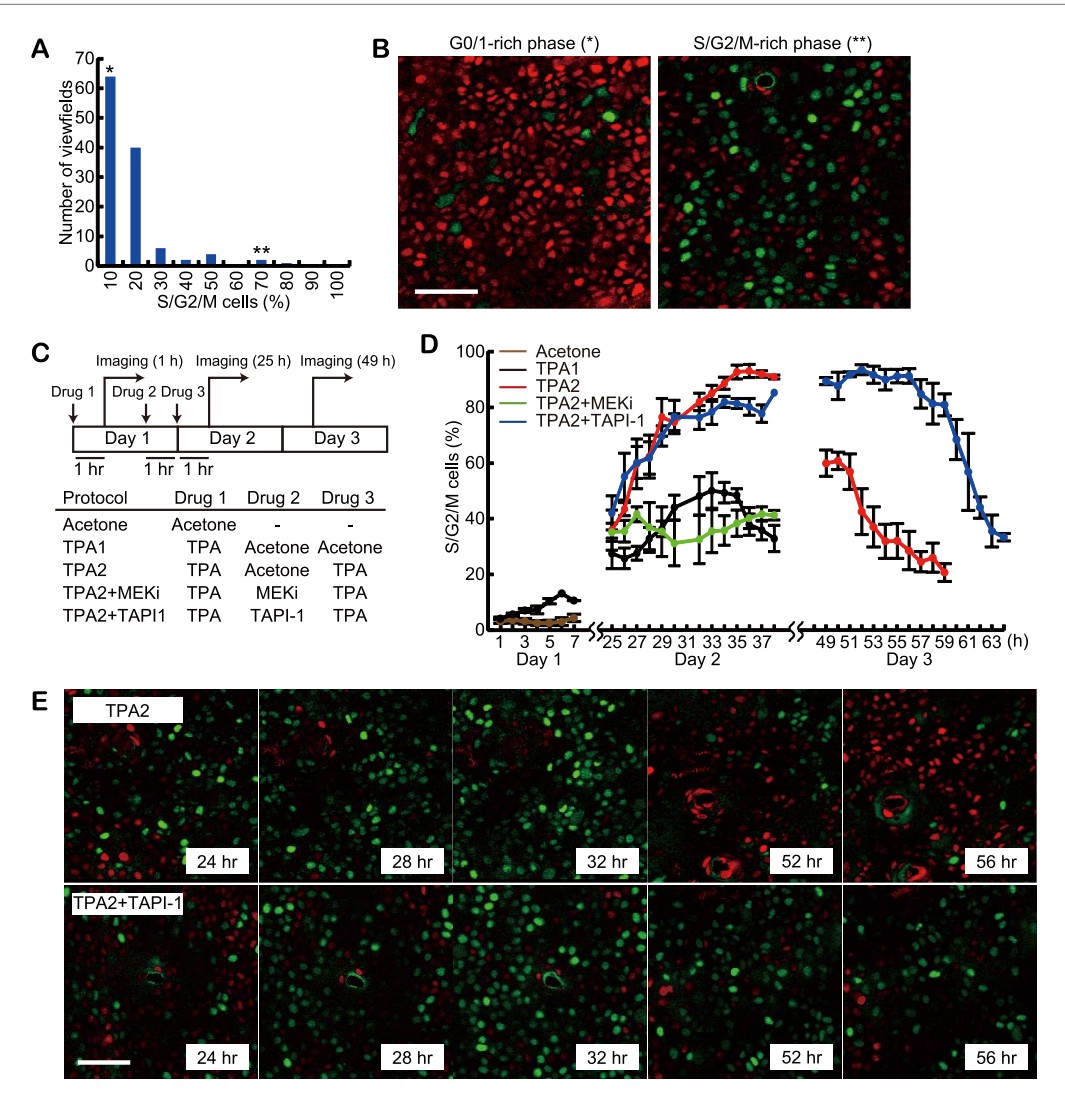

**Figure 6**. Delayed exit from S/G2/M phase by MMP inhibitor treatment. (**A**) Histogram of the percentage of S/G2/M cells in 120 viewfields (0.213 mm² each) of steady-state Fucci mice. About 700 cells were analysed for each viewfield. (**B**) Representative images of viewfields, where S/G2/M cells (green) were less than 10% (left, indicated by * in [**A**]) or more than 50% (right, indicated by ** in [**A**]). Red and green cells indicate G0/G1 and S/G2/M cells, respectively. Scale bar, 50 μm. (**C**) Protocols of double TPA treatments with or without inhibitors. Drugs were applied in the following concentrations: 0.5 nanomole TPA, 207 nanomole PD0329105, 2.0 nanomole TAPI-1, and 0.2 nanomole PD153035 in 20 μl acetone, or vehicle alone. For each protocol, at least three mice were observed. Images were acquired every 1 hr during 7- to 16-hr-imaging periods. Mice were returned to the cage after the imaging and the same area of the skin was revisited on the next day. (**D**) The proportion of S/G2/M cells. Average and S.E.M. are shown. At least three mice were used for each drug protocol. (**E**) Representative time lapse images of protocol TPA2 and TPA2 + TAPI-1. Note that the fraction of S/G2/M cells (green) was maintained at high level by TAPI-1 treatment (52 hr, 56 hr). Scale bar, 50 μm.

(*Kholodenko, 2006*; *Khavari and Rinn, 2007*). ERK plays a central role as a downstream target of a variety of growth signal pathways and its tight regulation in time, place, and degree of activation determine cell behaviours such as cell proliferation, migration, and differentiation (*Pouysségur et al., 2002*). Aberrant activation of ERK has been reported in many types of cancers especially by the oncogenic mutation of Ras and Raf (*Roberts and Der, 2007*). Despite the significance of ERK activity regulation, technical limitations have obscured how growth signal activation is generated and propagated throughout the tissue.

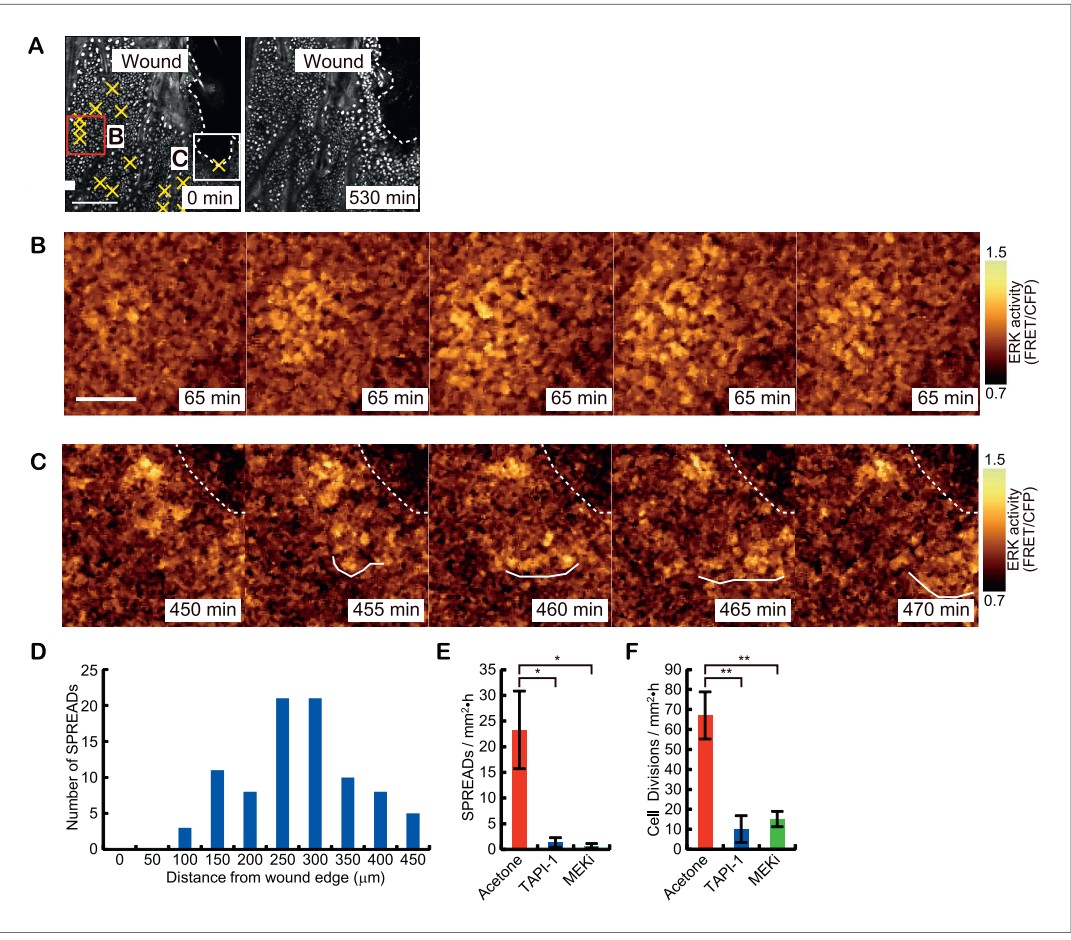

**Figure 7**. Two types of ERK activity propagation patterns during epithelial wound healing. (**A**) CFP images of the wounded ear skin of an Eisuke mouse. An epithelial wound was created on the ear skin of an Eisuke mouse 12 hr before observation. The wound edge and location of SPREADs observed during 9 hr of observation are shown by white dashed line and yellow crosses, respectively. (**B**) Time-lapse images of a representative SPREAD observed in the red-square region in (**A**). (**C**) Time-lapse images of a representative ERK propagation wave from the wound edge observed in the white-square region in (**A**). The images are cropped from *Video 3*. Scale bars are 100 μm (**A**) and 30 μm (**B** and **C**). (**D**) Histogram of the number of SPREADs classified by the distance from the wound edge. In total, 123 SPREADs were analysed from four independent experiments. (**E** and **F**) The frequencies of SPREAD (**E**) and cell division (**F**) per 1 hr in 1 mm². Eisuke mice were topically applied with the indicated drug at the time of wounding. Average and S.E.M. are shown. At least three mice were observed for each drug treatment. *p < 0.05; **p < 0.01 (paired Student's *t* test).

In this study, we exploited transgenic mice expressing a FRET biosensor of ERK and monitored ERK activity in epidermal cells at single cell resolution. Our long time lapse imaging of intact epidermis revealed radial propagation of ERK activity, where ERK activation originates from a few cells and propagate radially up to a radius of about 50 μm over 30 min until disappearance. We named this novel phenomenon as SPREAD and investigated its mechanism of ERK propagation and its role in cell cycle progression.

Although we previously reported stochastic propagation of ERK activity pulse in a single cell to its neighbouring cells (*Aoki et al., 2013*), the propagation was restricted within an array of a few cells rather than firework-like round propagation covering more than 100 cells. Our single cell analysis of SPREAD and inhibitor treatment revealed similarities and differences between the two patterns of ERK activation in vitro and in vivo. While they share in common the dependence of EGFRs and the production of their cognate ligands for ERK activation propagation, they differ in the efficiency and amplitude of ERK activation during propagation. Together with the difference in ERK

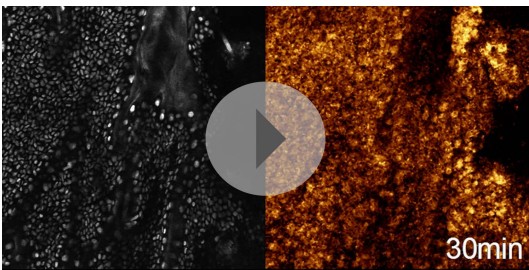

**Video 3**. SPREADs and ERK propagation waves in a wounded skin. A Time-lapse video of epidermis of an Eisuke mouse subjected to epithelial wounding 12 hr before imaging. The video length is about 9 hr (545 min). The video is shown by projection of five serial Z-stack images. SPREADs were observed at relatively distant areas from the wound edge while ERK activation waves are observed at the proximity of the wound edge. The FRET/CFP ratio is shown by gold pseudo-colours (right). Images were acquired every 5 min. The area of the viewfield is 0.213 mm². HF: hair follicle.

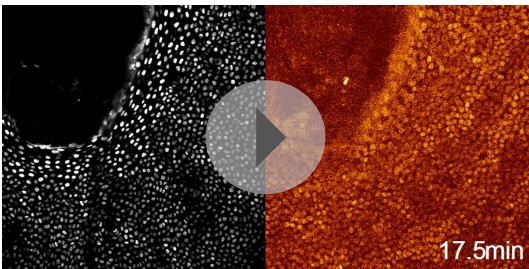

**Video 4**. ERK activation waves under MMP inhibitor treatment. A Time-lapse video of an Eisuke mouse subjected to epithelial wounding and 2.0 nanomole TAPI-1 treatment 12 hr before imaging. The FRET/CFP ratio is shown by gold pseudo-colours (right). Images were acquired every 5 min. The area of the viewfield is 0.213 mm².

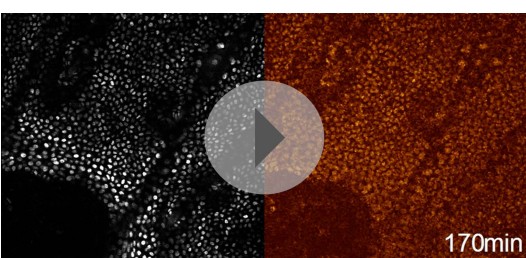

**Video 5**. ERK activation waves from the wound edge. A Time-lapse video of an Eisuke mouse subjected to epithelial wounding 12 hr before imaging. The video length is 87.5 min. This video was used for the analysis of ERK activation waves from the wound edge in *Figure 7*. The FRET/CFP ratio is shown by gold pseudo-colours (right). Images were acquired every 2.5 min. The area of the viewfield is 0.213 mm².

propagation manner between SPREADs in steady-state skin and ERK activation waves, this suggests that ERK propagation pattern might be altered depending on the surrounding environment of the cells, including the involvement of stem cells, cellular density, and pathological conditions. The induction of SPREADs by double TPA treatment raises the possibility that the alterations in SPREADs could be involved in tumorigenesis. Thus, it will be of special interest to observe SPREADs in various pathological conditions including cancer.

The dependency on EGFRs and MMPs strongly suggests that a common molecular mechanism operates between SPREADs and ERK activation propagation in tissue culture cells (*Aoki et al., 2013*). Unlike the ERK activation propagation in tissue culture cells, however, we could not conclude whether the inhibitors against EGFRs and MMPs suppress ERK activation in the initiator cell or abrogate intercellular propagation of the ERK activation. This is because we cannot clearly identify the ERK activation pulse at single cell resolution due partly to the long interval of image acquisition in vivo and low signal-to-noise ratio of in vivo FRET imaging.

As well as the established role of ERK activation in G1/S progression of cell cycle (*Torii et al., 2006*), the necessity of ERK to G2/M progression has also been reported in mammalian cells (*Tamemoto et al., 1992*; *Wright et al., 1999*). On the other hand, excessive ERK activation in G2 phase arrests cells in G2-phase. High ERK activity due to BRCA1 mutation, lack of Vaccinia H1-related (VHR) dual-specific protein tyrosine phosphatase, or stimulation by TPA induces G2-phase arrest (*Yan et al., 2005*; *Dangi et al., 2006*; *Rahmouni et al., 2006*; *Chambard et al., 2007*). Intriguingly, ERK activity is high around the G2/M phase in synchronized cells, but low when cells are arrested in M phase (*Torii et al., 2006*). These previous reports together with our findings show that exquisite regulation of ERK activation in terms of time, place, and degree is required for cell cycle progression and the localised transient activation of ERK in SPREAD can contribute to it. To validate the causal relationship between SPREADs and cell cycle progression, we need to develop methods to control cell cycle and/or ERK activity with the single cell resolution in the tissues.

We used the C57BL/6 strain and the FVB/N strain for the expression of the Fucci biosensor and the FRET biosensor, respectively. This is because bright fluorescence of melanin granules in C57BL/6 mice hampered quantification of

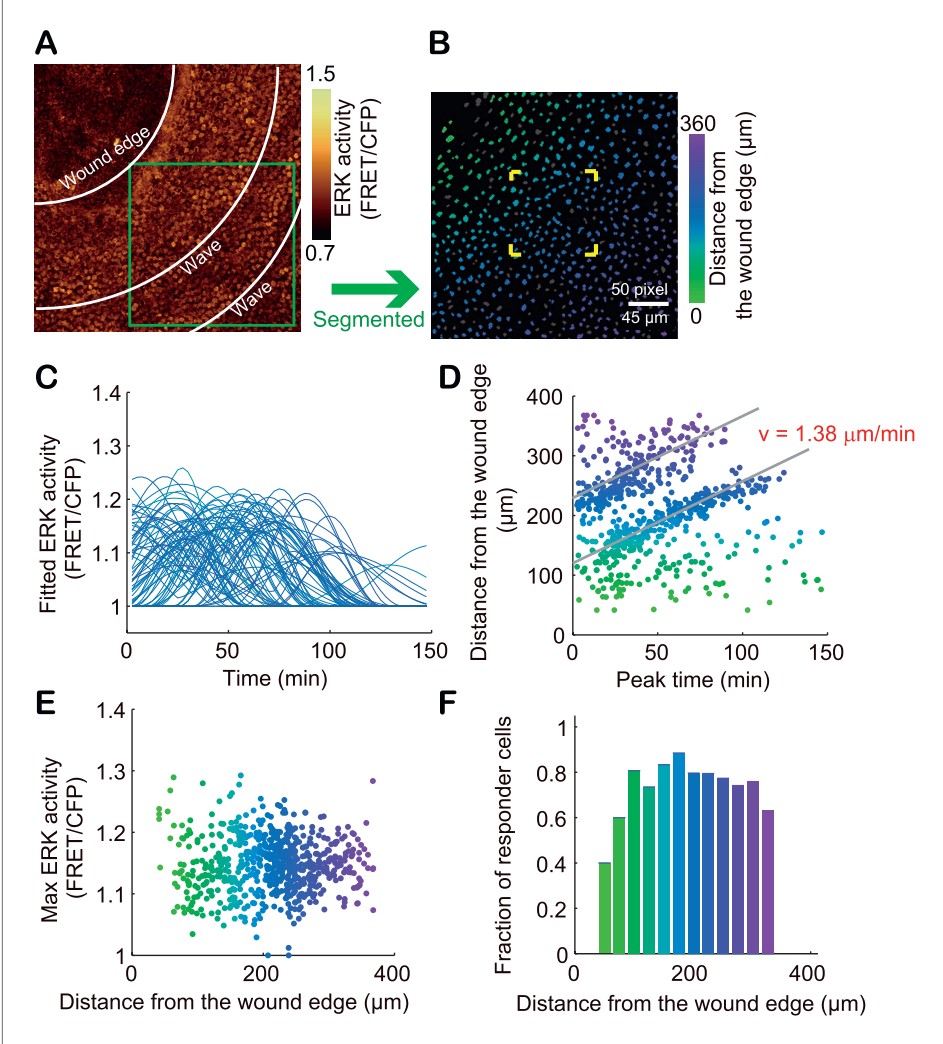

**Figure 8**. Single cell analysis of ERK propagation waves from the wound edge. (**A**) A gold pseudo-colour image of FRET/CFP ratio in an Eisuke mouse wounded 12 hr before observation. The image is cropped from **Video 5**. (**B**) The green-square region in (**A**) is magnified. The colours of segmented nuclei indicate the distance from the wound edge. Grey nuclei indicate non-responder cells. (**C**) With each nucleus, the time course of FRET/CFP values was approximated with a flat line or one to three sine curves as performed in **Figure 3**. (**D**) The distance from the wound edge to each cell was plotted against the peak ERK activation time to determine the velocity of ERK activation propagation. (**E**) The Max ERK activity of each cell was plotted against the distance from the wound edge. (**F**) The fraction of responder cells in each distance.

FRET signal and because Fucci mice with FVB/N background were not available. However, we speculate that the difference of the mouse strain will not affect our interpretation significantly by the following reasons. First, we observed significant increase of cell division in TPA-treated C57BL/6 mice, indicating that TPA promotes cell cycle in C57BL/6 as in FVB/N (**Figure 5**). Second, we observed SPREADs in C57BL/6 mice as in FVB/N, although quantitative data analysis was hampered by the fluorescence from the melanin granule.

In summary, here we revealed localized propagations of ERK activation regulate cell cycle progression in mouse epidermis. The propagation of ERK activity was achieved by cellular interaction via EGFRs and the production of their cognate ligands. Our finding of SPREAD achieved by in vivo imaging of dynamic changes in kinase activity pave a new way to understanding intrinsic relationship between cell cycle and the underlying growth signal in vivo regulated by an intercellular communications via EGFR ligands.

## Materials and methods

### Chemicals

TAPI-1 and PD0325901 were purchased from Calbiochem (San Diego, CA). PD0325901 and TPA were purchased from LC laboratories (Woburn, MA). PD153035 was purchased from MedChemexpress (Princeton, NJ).

### Mice

Transgenic mice expressing an ERK FRET biosensor EKAREV-NLS or EKAREV-NES were reported previously (*Kamioka et al., 2012*). Fucci mice expressing mAG-hGeminin (1/110) and mKO2-hCdt1 (30/120) were obtained from the Laboratory for Animal Resources and Genetic Engineering, RIKEN Center for Developmental Biology (*Sakaue-Sawano et al., 2008*). FRET mice were backcrossed for more than 10 generations to FVB/N before imaging. Fucci mice were bred in C57BL/6 background. Mice were housed in a specific-pathogen-free facility and received a routine chow diet and water ad libitum. To date, no disease or anomaly has been associated with the transgenic mice used in this study. 8- to 20-week-old mice were used for the in vivo imaging. The animal protocols were reviewed and approved by the Animal Care and Use Committee of Kyoto University Graduate School of Medicine (No. 10584).

### Microscopy

Two-photon excitation microscopy was performed with FV1200MPE-BX61WI upright microscopes, equipped with a 25X/1.05 water-immersion objective lens (XLPLN 25XWMP; Olympus, Tokyo, Japan) and an InSight DeepSee Ultrafast laser (0.95 W at 900 nm) (Spectra-Physics, Santa Clara, CA, USA). The laser power used for observation was 2–4% for mice expressing EKAREV-NES/NLS, 10–15% for Fucci mice. Scan speed was 8 μs/pixel. Images were recorded every 5 min for long-time imaging or every 1.5 min or 2.5 min for short-time imaging of Eisuke mice expressing EKAREV-NLS, and every 1 hr for imaging of Fucci mice and Eisuke mice expressing EKAREV-NES. The excitation wavelength was 840 nm for mice expressing a FRET biosensor and 910 nm for Fucci mice. For FRET mouse imaging, we used an IR-cut filter, RDM690 (Olympus), two dichroic mirrors, DM505 (Olympus) and DM570 (Olympus), and three emission filters, FF01-425/30 (Semrock, Rochester, NY) for second harmonic generation imaging, BA460-500 (Olympus) for CFP, and BA520-560 (Olympus) for FRET. For Fucci mouse imaging, we used an IR-cut filter, RDM690 (Olympus), two dichroic mirrors, DM505 (Olympus) and DM570 (Olympus), and three emission filters, FF01-472/30 (Semrock) for second harmonic generation images, BA495-540 (Olympus) for mAG, and BA575-630 (Olympus) for mKO2. Images were acquired with a viewfield of 0.213 mm$^2$ in 2–3 μm steps.

### In vivo imaging of ear skin and backskin

Mice were anesthetized with 1.5% isoflurane (Abbot Japan, Tokyo, Japan) inhalation and placed in the prone position on an electric heated stage maintained at 37°C. Mouse ear was sandwiched between a cover glass and a thermal conductive silicon gum sheet (*Figure 1B*). Mouse backskin was observed by a similar method (*Figure 1—figure supplement 2A*). Hairs were removed by depilation cream 24 hr before experiments. Mice were topically treated with 0.5 nanomole TPA, 2.0 nanomole TAPI-1, 0.2 nanomole PD153035, 207 nanomole PD0325901 in 20 μl acetone. Time-lapse images were acquired every 1.5 or 5 min for at least 9 hr, up to 29 hr, starting from 1 hr after the last treatment. For the wound healing experiments, an epithelial wound was created by a mini router No. 28600 (Kisopowertool MFG CO., LTD, Osaka, Japan) 12 hr before imaging. Time-lapse imaging was aborted when the body temperature and breathing conditions of mice deteriorated.

### Image processing

Acquired images were analysed with MetaMorph (Universal Imaging, West Chester, PA) as described previously (*Aoki and Matsuda, 2009*; *Kamioka et al., 2012*). Briefly, the level of FRET was shown by the FRET/CFP ratio image in intensity modulated display mode; eight colours from red to blue are used to represent the FRET/CFP ratio and the 32 grades of colour intensity are used to represent the signal intensity of the CFP image. The warm and cold colours indicate high and low FRET levels, respectively. For better presentation of SPREADs, a 3 × 3 median filter was applied to the FRET/CFP ratio images, and the resulting images were represented by a gold pseudo-colour. Both SPREADs and cell divisions were scored by visual inspections of time-lapse images. Projection images were

generated by averaging five serial FRET/CFP images for the analysis of wound images, in which epidermal basal cells appeared in different z planes due to invagination.

Cell cycle analysis was performed with Fucci mice. Melanin granules were excluded from the images based on the size and strong autofluorescence. For the identification of the nuclei of S/G2/M cells, images of mKO2-hCdt1 (30/120) were subtracted from images of mAG-hGeminin (1/110). The resulting images were processed with the segmentation function of the multi-dimensional motion analysis module of MetaMorph. Parameters used were: segmentation method, adaptive threshold; XY diameter, 4–10; local intensity above background, 30. The nuclei of G0/G1 cells were identified similarly.

## Analysis of SPREADs

The origins of SPREADs were initially determined by visual inspection and used in the following analysis. First, on the CFP image a square region was set around the origin of the SPREAD. In the selected region, nuclei were automatically recognized with a segmentation program. With each nucleus, the FRET/CFP ratio was calculated for all time frames. Second, the FRET/CFP values of each nucleus were smoothened by 15-min moving averages. The smoothed values in each cell were fit with a flat line first. If the coefficient for the difference between the experimental data and the approximated data is less than 0.003, the cell was defined as a non-responder. Otherwise, values were fit with increasing number of sine curves until the coefficient became less than 0.003. The fitting was ended if the number of sine curves reached three. Cells fit by sine curve(s) were defined as responder cells. For the responder cells, the approximated sine curves were used to obtain the following parameters; peak time, the time when the FRET/CFP ratio reaches the zenith; amplitude, the FRET/CFP ratio at the peak time; duration, the duration during which the FRET/CFP ratio was higher than 50% of the peak ERK activity. Third, for each nucleus of the responder cells, the distance from the origin of the SPREAD was plotted against the peak time. The location of SPREAD origin was optimised by changing the ordinates of the SPREAD origin between −10 and 10 pixels and finding the position that gave the highest correlation efficiency between the data and the linear regression curve. Fourth, circles with increasing radius were drawn and the fraction of responder cells therein was obtained. When the fraction of responder cells became less than 50%, the obtained value was defined as the radius of the SPREAD. The data analysis was performed with MATLAB (MathWorks, Natick, MA).

## Monte Carlo simulation of SPREAD distribution

A statistical test was performed with the null hypothesis that SPREAD locations are randomly distributed. A test statistic was the average distance between each SPREAD centre and its nearest neighbour's centre. With the number of SPREADs and analysed area of the data set of mouse #102,100,000 trials of Monte Carlo simulations were performed to obtain the histogram of the expected distances on the null hypothesis. p-value was calculated based on the histogram with one-side test.

## Statistical analysis

Data are expressed as mean ± S.E.M. except for *Figure 3*, where we used S.D. An unpaired Student's *t* test was used to evaluate statistically significant differences. Data analysis was performed using Excel software. *p < 0.05; **p < 0.01; ***p < 0.001.

## Acknowledgements

We thank Y Inaoka, K Hirano, R Tabata, and A Kawagishi for technical assistance, Y Manabe for instruction on epithelial wounding with a router, James Hejna for helpful discussion. This research was supported by a Grant of Platform for Dynamic Approaches to Living System and a Grant-in-Aid for Scientific Research on the Innovative Area 'Fluorescence Live imaging' (No. 22113002) to MM and by a Grant-in-Aid for Japan Society for the Promotion of Science Fellows to HT and YF, from the Ministry of Education, Culture, Sports, Science, and Technology of Japan.

## Additional information

### Funding

| Funder | Grant reference number | Author |
| --- | --- | --- |
| Ministry of Education, Culture, Sports, Science, and Technology | | Michiyuki Matsuda |

| Funder | Grant reference number | Author |
|---|---|---|
| Ministry of Education, Culture, Sports, Science, and Technology | No. 22113002 | Michiyuki Matsuda |
| Japan Society for the Promotion of Science | Graduate Student Fellowship | Toru Hiratsuka, Yoshihisa Fujita |

The funders had no role in study design, data collection and interpretation, or the decision to submit the work for publication.

## Author contributions

TH, Conception and design, Acquisition of data, Analysis and interpretation of data, Drafting or revising the article; YF, Analysis and interpretation of data, Drafting or revising the article; HN, KA, Conception and design, Analysis and interpretation of data; YK, Transgenic mice; MM, Conception and design, Analysis and interpretation of data, Drafting or revising the article

## Ethics

Animal experimentation: The animal protocols were reviewed and approved by the Animal Care and Use Committee of Kyoto University Graduate School of Medicine (No. 10584).

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
