## [Decision Letter]

Thank you for sending your work entitled “Waves of localized extracellular signal-regulated kinase activation regulate cell cycle progression in mouse epidermis” for consideration at *eLife*. Your article has been favorably evaluated by Fiona Watt (Senior editor) and 4 reviewers, two of whom are members of our Board of Reviewing Editors. Fiona Watt declared a potential conflict of interest and therefore has not had input into this decision letter, except at the proofing stage.

The Reviewing editor and the other reviewers discussed their comments before we reached this decision, and the Reviewing editor has assembled the following comments to help you prepare a revised submission.

This study builds upon previous work showing that ERK activation can propagate at constant speed from cell to cell in tissue culture, that the spreading requires metaloproteinase activity, and that the amplitude of the ERK activation does not decrease as the distance traveled by the activation wave increases. Here the authors set out to test whether this cell culture work is relevant to the situation in mouse skin in vivo by making use of the Eisuke mouse, which stably expresses an ERK reporter.

They report two types of ERK activity propagation in mouse skin. The first (which they refer to by the acronym SPREAD) begins stochastically, like the waves in vitro, but decreases in amplitude and fizzles out over a distance of ∼50 µm. The second occurs in response to wounding, and is characterized by repeated, self-organizing wave fronts of ERK activation that do not decrease in amplitude. Both of these types of waves depend upon metalloprotease activity and MEK activation. It is not yet clear what makes the SPREAD waves peter out vs. the more sustained propagation of the wound-response waves.

This is a beautiful paper, using powerful quantitative methods to discover interesting phenomena. The discovery of the wound-response waves is particularly exciting because of the high likelihood they are physiologically and functionally important.

Major comments:

1) The authors should make clear that there are two different types of propagation described in this report: what they call SPREAD, which is a sporadic burst of ERK activation in normal skin and TPA-treated skin; and a more wave-like, longer distance response to wounding. For example, the title of the manuscript refers to waves, but this is only observed in the wounding paradigm. The manuscript would be improved by placing the analysis within overall context of diffusive spatial propagation vs. trigger wave spatial propagation. This would help to put these particular examples of intercellular signaling into a broader context.

2) There is significant heterogeneity of SPREADs and cell division amongst the mice that were analysed (Figure 4). Might this be due to variations in basal levels of ERK signaling between different mice, or is it related to age, sex or other factors? If the same mouse was repeatedly analysed, would the number of SPREADs/cell divisions vary significantly each time? Are mice that display fewer SPREADs under basal conditions equally responsive to TPA? Can the authors comment on this or provide any insight?

3) The authors state in the Results section that their observations show that the propagation of ERK activation in SPREADs is mediated by EGFR-ligands released by MMPs. However, their data shows that the EGF and MMP inhibitors decrease the number of SPREADs, which may not necessarily be the same as blocking the propagation of ERK activation (Figure 5). This should be clarified.

4) The cell cycle analysis is performed using Fucci mice without the ERK sensor. Therefore, the argument that SPREADs play an important role in G2/M progression makes the presumption that SPREADs occur in Fucci mice similar to the Eisuke mice. These mice are backcrossed to different strains. This caveat should be mentioned in the text. Moreover, the data that support an association between SPREADs and cell cycle regulation are weak. The authors find that mice with frequent SPREADs also have more frequent cell division, but these events do not appear to have a definable temporal relationship. The authors should comment on this.

5) The authors show data to support a spatial relationship (Figure 4) between SPREADs and cell division. However, the data in Figure 3 demonstrate that SPREADs have an average radius of ∼ 47 uM, but in Figure 4, the authors use 100 uM from the SPREAD origin to define a cell as “within the SPREAD area”. These data should be recalculated using values described for SPREADs in Figure 3, or a justification should be provided for why 100 uM was used.

6) Figure 7–figure supplement 1 is important enough that it should be in the main manuscript.

---

## [Author Response]

*1) The authors should make clear that there are two different types of propagation described in this report: what they call SPREAD, which is a sporadic burst of ERK activation in normal skin and TPA-treated skin; and a more wave-like, longer distance response to wounding. For example, the title of the manuscript refers to waves, but this is only observed in the wounding paradigm. The manuscript would be improved by placing the analysis within overall context of diffusive spatial propagation vs. trigger wave spatial propagation. This would help to put these particular examples of intercellular signaling into a broader context*.

We believe that the wavefront during wound healing is generated by the superposition of a number of ERK activation pulses evoked in close temporal and spatial proximity. Similarity in the velocity of ERK activity propagation and the dependence on MMP supports this proposal. Therefore, we would like to refrain from emphasizing the difference between the two types of ERK activity propagation and have included a statement, in the Results section, that ERK activation propagation can be synchronized and superimposed to generate trigger waves if a number of ERK activation pulses are evoked in close temporal and spatial proximity.

We admit that “wave” in the title was misleading. We did not intend to refer to the ERK activity propagation from the wound edge. Furthermore, by the suggestion, we have toned down not to conclude the function of SPREADs in the title. The revised title is as follows: “Intercellular propagation of extracellular signal-regulated kinase activation revealed by in vivo imaging of mouse skin.”

*2) There is significant heterogeneity of SPREADs and cell division amongst the mice that were analysed (*Figure 4*). Might this be due to variations in basal levels of ERK signaling between different mice, or is it related to age, sex or other factors? If the same mouse was repeatedly analysed, would the number of SPREADs/cell divisions vary significantly each time? Are mice that display fewer SPREADs under basal conditions equally responsive to TPA? Can the authors comment on this or provide any insight*?

Heterogeneity of SPREADs vs. age etc.: these are very important questions that we wished to answer. Unfortunately, we have not found any correlation of the SPREAD frequency with sex, age, or other factors. The number of mice used in our study may not be sufficient to find the conditions for the induction of SPREADs. For the benefit of readers, we have included the sex and age of each mouse in Figure 4—figure supplement 1 and Figure 4—figure supplement 2.

Heterogeneity of SPREADs vs. basal ERK activity: we compared the basal ERK activities between mice with frequent SPREAD (#102, 107, 112, and 115) and mice with infrequent SPREAD (#113, 114, and 120). The average ERK activity was higher in SPREAD-frequent mice than SPREAD-infrequent mice; however, the difference was not statistically significant by student t-test (P=0.14). We have stated this observation in the Results section and included the data as Figure 4—figure supplement 1.

Heterogeneity of SPREADs in each mouse: we have not succeeded in the second long-term imaging in the same mouse. The long-term in vivo imaging, which is required for the observation of the SPREAD-frequent phase, is still challenging. However, it could become possible in the near future with better care of anaesthetized mice during live imaging.

*3) The authors state in the Results section that their observations show that the propagation of ERK activation in SPREADs is mediated by EGFR-ligands released by MMPs. However, their data shows that the EGF and MMP inhibitors decrease the number of SPREADs, which may not necessarily be the same as blocking the propagation of ERK activation (*Figure 5*). This should be clarified*.

We have to admit that the molecular mechanism of SPREAD has not been extensively characterized due to technical difficulties. However, the resemblance between SPREAD and ERK activation pulses in vitro strongly suggests that the same mechanism operates in these two phenomena. We have included this point in the Discussion section.

*4) The cell cycle analysis is performed using Fucci mice without the ERK sensor. Therefore, the argument that SPREADs play an important role in G2/M progression makes the presumption that SPREADs occur in Fucci mice similar to the Eisuke mice. These mice are backcrossed to different strains. This caveat should be mentioned in the text. Moreover, the data that support an association between SPREADs and cell cycle regulation are weak. The authors find that mice with frequent SPREADs also have more frequent cell division, but these events do not appear to have a definable temporal relationship. The authors should comment on this*.

We agree to the reviewers’ criticism that the use of different strain may affect the interpretation of the results. We established the FVB/N^EKAREV-NLS^ mice to avoid the fluorescence of melanin granules in the skin of C57BL/6 ^EKAREV-NLS^ mice. In our preliminary experiments, however, we observed SPREADs in C57BL/6 ^EKAREV-NLS^ mice as in FVB/N^EKAREV-NLS^ mice. Therefore, we do not think the difference of mouse strain will significantly affect the results. We have discussed this issue in the Discussion section. We also admit that, without techniques to control cell cycle or ERK activity in vivo, the causal relationship between SPREAD and cell cycle cannot be validated. We have toned down our conclusion in the Title and Abstract, and commented these points in the Discussion section.

*5) The authors show data to support a spatial relationship (*Figure 4*) between SPREADs and cell division. However, the data in*
Figure 3
*demonstrate that SPREADs have an average radius of ∼ 47 uM, but in*
Figure 4*, the authors use 100 uM from the SPREAD origin to define a cell as “within the SPREAD area”. These data should be recalculated using values described for SPREADs in*
Figure 3*, or a justification should be provided for why 100 uM was used*.

We defined the radius of a SPREAD as the distance from the centre to the inflection point of the histogram of the responder cells (Figure 3). Therefore, the ERK activation wave actually reaches further away. Based on the data shown in Figure 3 and Figure 3—figure supplement 1, we analysed the cells within 100 um from the SPREAD centre. We have referred to this issue in the Results section.

*6) Figure 7–figure supplement 1 is important enough that it should be in the main manuscript*.

We have renamed Figure 7–figure supplement 1 to become Figure 8.